 

# SPRTN is a mammalian DNA-binding metalloprotease that resolves DNA-protein crosslinks

Jaime Lopez-Mosqueda[1], Karthik Maddi[1,2†], Stefan Prgomet[1†], Sissy Kalayil[1,2], Ivana Marinovic-Terzic[3], Janos Terzic[3], Ivan Dikic[1,2*]

[1]Institute of Biochemistry II, Goethe University School of Medicine, Frankfurt, Germany; [2]Buchmann Institute for Molecular Life Sciences, Goethe University, Frankfurt, Germany; [3]Department of Immunology and Medical Genetics, University of Split, School of Medicine, Split, Croatia

**Abstract** Ruijs-Aalfs syndrome is a segmental progeroid syndrome resulting from mutations in the *SPRTN* gene. Cells derived from patients with SPRTN mutations elicit genomic instability and people afflicted with this syndrome developed hepatocellular carcinoma. Here we describe the molecular mechanism by which SPRTN contributes to genome stability and normal cellular homeostasis. We show that SPRTN is a DNA-dependent mammalian protease required for resolving cytotoxic DNA-protein crosslinks (DPCs)— a function that had only been attributed to the metalloprotease Wss1 in budding yeast. We provide genetic evidence that SPRTN and Wss1 function distinctly in vivo to resolve DPCs. Upon DNA and ubiquitin binding, SPRTN can elicit proteolytic activity; cleaving DPC substrates and itself. *SPRTN* null cells or cells derived from patients with Ruijs-Aalfs syndrome are impaired in the resolution of covalent DPCs in vivo. Collectively, SPRTN is a mammalian protease required for resolving DNA-protein crosslinks in vivo whose function is compromised in Ruijs-Aalfs syndrome patients.

*For correspondence: dikic@biochem2.uni-frankfurt.de

†These authors contributed equally to this work

## Introduction

DNA is continuously subjected to various forms of damage stemming from endogenous and exogenous sources. The DNA lesions are highly diverse and can include UV-induced photoproducts, inter- and intra-strand crosslinks, abasic sites, oxidative damage, bulky chemical lesions or single and double strand breaks (*Hoeijmakers, 2009*). Specialized DNA repair systems have evolved to repair the multitude of lesions and these repair systems collectively contribute to maintaining genome stability and cellular survival (*Jackson and Bartek, 2009*). Numerous proteins assemble on DNA to carryout essential metabolic transactions and their recruitment and removal is precisely regulated.

Naturally occurring cellular processes, such as histone demethylation, can also lead to covalent DNA-protein crosslinks (DPCs) of neighboring proteins (*Shi et al., 2004*; *Tsukada et al., 2006*). DPCs, if not resolved, are formidable impediments to ongoing DNA replication forks, gene transcription (*Stingele and Jentsch, 2015*) and conceivably chromatin remodeling. As DNA replication machinery synthesizes new DNA at the replication fork, DNA supercoiling is created ahead of and behind the replication fork, profoundly altering DNA topology. Topoisomerases 1 and 2 (Top1 and Top2) relax DNA supercoiling by breaking and re-ligating one or both strands of the DNA, respectively (*Pommier and Marchand, 2012*). Top1 and Top2 naturally form DPCs as part of an obligate intermediate step required for the re-ligation step. Because enzymatic reactions have an intrinsic error rate, malfunction in the re-ligation steps cause Top1 and Top2 to be permanently trapped on DNA in the form of DPCs. DPCs must be adequately resolved to not only ensure the

timely completion of chromosome duplication and segregation but also for the synthesis of proteins that are necessary for cellular homeostasis. Yet, the molecular mechanisms by which DPCs are repaired remain poorly characterized.

Recently, a repair mechanism that resolves DPCs was identified in the yeast, *Saccharomyces cerevisiae* (*Balakirev et al., 2015*; *Stingele et al., 2014*). Wss1 was shown to counteract DPCs induced by the anti-cancer agent Camptothecin (CPT) or those induced by the unspecific crosslinking agent, formaldehyde. Moreover, yeast cells missing key components of the DPC repair system succumb to cell death upon exposure to CPT, underscoring the importance of DPC removal. In *Xenopus laevis* egg extract, an analogous DPC repair mechanism has been identified in vitro (*Duxin et al., 2014*), however the identity of the protease(s) involved was not elucidated. The recent identification of a DPC-repair pathway suggests that DNA-protein crosslink resolution is an important repair mechanism that is conserved throughout evolution. However, it is unknown whether a protease that counteracts DPCs exists in mammalian cells.

DNA is an important molecule that is targeted for time-dependent deterioration. This is highlighted in the rare inherited disorders called segmental progeroid syndromes in which genome maintenance is compromised and select indicators of physiological aging are accelerated (*Lopez-Otin et al., 2013*). SPRTN (also known as Dvc1 or *C1orf124*) was initially discovered as being an important protein for translesion DNA synthesis (*Centore et al., 2012*; *Davis et al., 2012*; *Ghosal et al., 2012*; *Juhasz et al., 2012*; *Kim et al., 2013*; *Machida et al., 2012*; *Mosbech et al., 2012*). SPRTN is a multi-domain containing protein that is endowed with an SprT-like domain, a SHP and PIP box as well as a ubiquitin binding zinc finger (UBZ) domain. SPRTN binding to ubiquitin and to PCNA was suggested to be critical for its recruitment to sites of UV-induced DNA damage, while interactions with a ubiquitin-dependent molecular segregase p97 (known as VCP in higher eukaryotes and Cdc48 in yeast) are implicated in extraction of the translesion polymerase Pol eta (*Davis et al., 2012*; *Mosbech et al., 2012*). Based on the presence of an SprT-like metalloprotease domain with similarity to the WLM domain in Wss1, it was proposed that SPRTN and Wss1 could belong to a related family of DPC-resolving proteases (*Stingele et al., 2015*). Later work revealed that SPRTN is an essential protein, as mice with homozygous null alleles die very early during embryonic development (*Maskey et al., 2014*). Furthermore, hypomorphic murine *SPRTN* alleles render mice with premature aging phenotypes (*Maskey et al., 2014*). Very recently, we identified rare *SPRTN* germ-line mutations in three patients from two unrelated families afflicted with Ruijs-Aalfs syndrome (*Lessel et al., 2014*). The first proband inherited homozygous point mutations that resulted in a premature stop codon and subsequent loss of the carboxyl-terminal half of the SPRTN protein (herein referred to as SPRTN-ΔC). The second set of probands inherited biallelic mutations such that one allele was also a point mutation that, coincidentally, is highly reminiscent to the SPRTN-ΔC, while the second allele has a tyrosine to cysteine substitution at position 117 (herein referred to as SPRTN-Y117C). Individuals harboring these mutations were shown to have segmental progeroid features and developed early onset hepatocellular carcinoma (*Lessel et al., 2014*; *Ruijs et al., 2003*). In addition, cells derived from affected individuals showed genomic instability associated with DNA replication stress. DNA fiber assays confirmed slower replication fork movements in cells from affected individuals and the slowing of replication forks was further exacerbated with the replication inhibitor Aphidicolin (*Lessel et al., 2014*), suggesting that cells with SPRTN mutations had difficulty replicating through blocking lesions.

In this report we identify SPRTN as the first mammalian protease that is required for the resolution of DPCs in cells. Cells deficient in SPRTN expression or cells from individuals with SPRTN mutations are sensitive to formaldehyde, CPT and Etoposide, treatments that are known to induce covalent DNA-protein crosslinks. Moreover, we show that drug-induced DPCs are not resolved and are sustained in cells lacking SPRTN proteolytic activity. In vitro, SPRTN elicits proteolytic activity as it can readily act on itself and other DNA binding proteins. Finally, we provide molecular insight into the defects in patients afflicted with Ruijs-Aalfs syndrome and show that DPC resolution is compromised in these individuals rendering them susceptible to premature aging and cancer predisposition.

## Results

### SPRTN reconstitution in yeast

Since the recent discovery of a dedicated DPC-resolution mechanism in the budding yeast, there has been much speculation that SPRTN is the mammalian orthologue of the yeast protease Wss1. Although SPRTN and Wss1 share some common domains like a zinc metalloprotease domain and a SHP/VIM domain (p97/Cdc48 binding), they are also distinct in that they recognize substrates differently. SPRTN preferentially binds to ubiquitin via its UBZ domain whereas Wss1 binds to SUMO via its SIM motives (*Figure 1A*) (*Balakirev et al., 2015*; *Centore et al., 2012*; *Davis et al., 2012*; *Ghosal et al., 2012*; *Juhasz et al., 2012*; *Machida et al., 2012*; *Mosbech et al., 2012*; *Stingele et al., 2014*). Despite the similarities in domain organization, the overall amino acid sequence conservation between SPRTN and Wss1 is relatively poor, particularly within the protease domains (*Figure 1B*), sharing only the conserved signature HEXXH motif found in the SprT family of metalloproteases, with glutamic acid being the catalytic amino acid. Recently, it was shown that Wss1 provides resistance to CPT in the absence of Tdp1, a tyrosyl phosphodiesterase that removes Top1 remnants from 3′ ends of DNA (*Liu et al., 2002*; *Pommier et al., 2014*; *Pouliot et al., 1999*), such that Wss1 and Tdp1 double deletion mutant strains die upon exposure to the DPC-inducing drug CPT (*Balakirev et al., 2015*; *Stingele et al., 2014*). This experimental setting provided an opportunity to directly test whether SPRTN can functionally complement Wss1 activity and rescue the CPT-dependent lethality observed in *wss1Δ tdp1Δ* cells. Towards this aim, we expressed SPRTN or mutants thereof under the control of the endogenous Wss1 promoter. It is noteworthy that *wss1Δ tdp1Δ* cells accumulate spontaneous second-site mutations that often suppressed the CPT-induced lethality. To circumvent this complication, we opted for a dual protein depletion system where Wss1 transcription is under control of a tetracycline repressible promoter and Wss1 itself is reduced by N-end rule degradation (*Bachmair et al., 1986*; *Gnanasundram and Kos, 2015*) (*Figure 1—figure supplement 1*). Using this system, Wss1 depletion occurred as rapidly as 40 min after doxycycline treatment (*Figure 1—figure supplement 1*) and without overt consequences to the overall fitness of the cells. Consistent with previous reports, *wss1Δ tdp1Δ* cells lost viability when exposed to CPT (*Mullally et al., 2015*; *Stingele et al., 2014*). Surprisingly, expression of SPRTN could not support the growth of *wss1Δ tdp1Δ* cells on CPT containing media (*Figure 1C*). *SPRTN* also failed to complement Wss1 activity when SPRTN protein levels was increased to high levels using the inducible Galactose promoter (*Figure 1—figure supplement 2*). The lack of functional complementation was not simply due to poor induction of SPRTN expression or a failure of SPRTN to reach the nucleus, as SPRTN expression is readily detected by Western blot analysis and its expression co-localizes with DAPI nuclear stain (*Figure 1—figure supplement 2*). Because SPRTN failed to functionally complement Wss1 protease activity, we could not fully characterize SPRTN-Y117C or SPRTN-ΔC in this experimental setting. We did observe, however, that ectopic SPRTN expression induced a slow-growth phenotype, in a catalytic-dependent manner, such that neither SPRTN-E112A nor SPRTN-Y117C expression induced this slow-growth phenotype in *tdp1Δ* cells. This genetic assay suggests that SPRTN-E112A and SPRTN-Y117C are catalytically inactive in yeast and that SPRTN expression elicits a dominant negative function over endogenous Wss1. Taken together, we conclude that SPRTN cannot functionally complement Wss1 function due to either additional regulation of SPRTN not present in yeast or to inherently different pathways between yeast and metazoan cells involved in DPC resolution.

### DPCs are removed in a SPRTN dependent-manner

To implicate SPRTN as the mammalian protease that plays a critical role in the resolution of DPCs, we utilized murine embryonic fibroblast cells (MEFs) derived from conditional knockout SPRTN mice (*SPRTN$^{flox/-}$;Cre-ER$^{T2}$*) (*Maskey et al., 2014*) and tested whether SPRTN expression allows for resistance to DPC-inducing drugs. These conditional *SPRTN* MEFs become *SPRTN* knock-out (*SPRTN-KO*) after 48 hr of 4-hydroxytamoxifen (4-OHT) treatment and these cells will eventually decline in culture after 6–7 days (*Maskey et al., 2014*). Using these cells, we performed clonogenic survival assays after acute formaldehyde and CPT treatment, as well as chronic Etoposide treatment. We found that *SPRTN-KO* cells were more sensitive to formaldehyde, CPT, and Etoposide treatments than control MEFs (*Figure 2A*).



**Figure 1.** SPRTN fails to complement Wss1 activity in yeast. (**A**) Comparison of SPRTN and Wss1 domain organization. (**B**) Comparison of SprT-like domains from selected SPRTN-containing organisms. Amino acid highlighted in red corresponds to the tyrosine mutated in patients with Ruijs-Aalfs syndrome (SPRTN-Y117C), identical amino acids are in blue and similar amino acids are in yellow. The catalytic glutamic acid and zinc-coordinating histidines are indicated by bold text. (**C**) Yeast spot assay. Five-fold dilutions of wildtype, *tdp1Δ*, and *wss1Δ tdp1Δ* cells. *wss1Δ tdp1Δ* cells harboring empty plasmid (— = pRS415) or plasmid encoding wild-type *SPRTN* or corresponding mutants under the endogenous Wss1 promoter. Cells were spotted on CSM-LEU plates with 2 ug/mL doxycycline or 2 ug/mL doxycycline and 40 uM CPT. Doxycycline is used to acutely deplete Wss1 protein levels in *tdp1Δ* cells to effectively obtain *wss1Δ tdp1Δ* cells.

The following figure supplements are available for figure 1:

**Figure supplement 1.** Dual Wss1 protein depletion system.

**Figure supplement 2.** SPRTN complementation in yeast.



**Figure 2.** SPRTN is required for DPC removal. (**A**) Clonogenic assay of control *SPRTN,* and *SPRTN*-KO MEFs. Control *SPRTN* MEFs were treated with 4-OHT for 24 hr to induce CRE-mediated excision of SPRTN. Cells were then treated with formaldehyde, Etoposide or CPT at the indicated drug concentrations for 24 hr (72 hr for Etoposide) and subsequently allowed to form colonies for 10 days. Cells were plated in triplicate. Error bars represent S.E.M. (**B**) In vivo complex of Enzyme (ICE) assay. Schematic illustration of the method used to isolate Etoposide-induced covalent Top2 DNA-protein

*Figure 2 continued on next page*

*Figure 2 continued*

crosslinks from cells. (**C**) ICE assay of control *SPRTN* MEFs, *SPRTN-KO* MEFs, and conditional *SPRTN* MEFs reconstituted with *SPRTN, SPRTN-E112A, SPRTN-Y117C, or SPRTN-ΔC.* Cells were either untreated or treated for 30 min with 25 uM Etoposide and immediately processed for ICE assay or processed 1 and 2 hr after washing out Etoposide (recovery). 1 ug of DNA was loaded for each time-point per cell line. Membrane was probed with an antibody against Top2 and subsequently re-probed with anti-thymine dimer antibody after 1 min UV treatment to control for DNA loading. Representative Western blot indicating SPRTN levels after reconstitution of *SPRTN-KO* MEFs.

To functionally dissect SPRTN's role in providing resistance to DPC-inducing drugs, we sought to directly assess the prevalence of DPCs in *SPRTN-KO* cells. We took advantage of the fact that Etoposide induces Top2 covalent complexes (Top2cc) on DNA (*Wu et al., 2011*) and that these DPCs can be detected using the in vivo complex of enzyme (ICE) assay (*Trask and Muller, 1988*). Briefly, the ICE assay depends on the separation of Top2 covalently bound to DNA from free Top2 by ultracentrifugation through a cesium chloride gradient. DNA will pellet along with Top2 covalently bound to DNA, while free Top2 will remain in suspension near the top of the centrifuge tube (*Figure 2B*). Equal amounts of isolated DNA are subsequently dot-blotted onto nitrocellulose filters. The presence of Top2cc can be detected with specific antibodies raised against Top2 subunits. Using this experimental approach, Top2 DPCs were not appreciably detected in untreated cells but were detected in DNA isolated from control SPRTN cells treated with Etoposide for 0.5 hr (*Figure 2C*), which is consistent with previous reports (*Subramanian et al., 1995*). The Top2 signal diminished 1 hr after Etoposide treatment, indicating that Top2 had been cleared from DNA. In *SPRTN-KO* MEFs, Top2 signals are also detected at 0.5 hr into the Etoposide treatment. In contrast to control MEFs, Top2 signals are sustained throughout the duration of the time course in *SPRTN-KO* MEFs (*Figure 2C*). These results indicate that Top2 is not proteolyzed in cells lacking the SPRTN protease. To further substantiate our findings, we reconstituted conditional *SPRTN-KO* MEFs with human *SPRTN* (*Figure 2C*) or mutant alleles and analyzed Top2cc resolution using the ICE assay. As expected, knockout MEFs reconstituted with SPRTN-WT are able to resolve Top2cc from DNA, albeit with slower kinetics than the control *SPRTN* MEFs probably due to low levels of ectopic SPRTN expression. Contrary to SPRTN-WT, *SPRTN-KO* MEFs reconstituted with *SPRTN-ΔC* were delayed in Top2cc resolution under these conditions. Notably, *SPRTN-KO* MEFs could not be stably reconstituted with either *SPRTN-E112A* or *SPRTN-Y117C*. Nevertheless, SPRTN-KO MEFs reconstituted with either SPRTN-E112A or SPRTN-Y117C by transient transfection, failed to resolve Top2 DPCs. This finding is consistent with the notion that these alleles code for mutant SPRTN proteins lacking essential proteolytic activity. These results strongly suggest that SPRTN is the mammalian protease responsible for DPC removal and that this function is compromised in cells expressing SPRTN-Y117C, SPRTN-E112A or SPRTN-ΔC mutant proteins.

## SPRTN is a DNA binding protease in vivo and in vitro

SPRTN contains an SprT-like domain within its amino-terminus with the characteristic HEXXH catalytic signature motif found in zinc-dependent metalloproteases (*Figure 1B*). However, SPRTN protease activity has not been reported. Consistent with SPRTN being a protease, specific SPRTN cleavage fragments can be detected by Western blot analysis or live-cell imaging when SPRTN is over-expressed in cells (*Figure 3A and B* and *Figure 3—figure supplement 1A and B*). Importantly, the SPRTN fragments are not detected when the catalytic glutamic acid in the HEXXH catalytic motif is mutated to alanine (E112A). Interestingly, a similar attenuation of proteolytic fragments is observed when cells express the SPRTN-Y117C or SPRTN-ΔC (*Figure 3A*).

To determine if SPRTN can proteolyze DNA binding proteins, we performed an unbiased protease screen for SPRTN substrates using mass spectrometry to directly detect modification of newly formed amino-termini resulting from proteolytic cleavage of candidate proteins. In this screen we found that histones were significantly over-represented as candidate SPRTN substrates. It is likely that core histones can be relevant SPRTN substrates as they are intimately associated with DNA and are poised to become crosslinked upon exposure to endogenous formaldehyde. Consistent with this initial finding, we can observe the presence of a clear histone H3 fragment when SPRTN is overexpressed in 293T-HEK cells (*Figure 3B*). Importantly, the histone H3 cleavage product is not detected when SPRTN-E112A is over-expressed irrespective of stimulation with Etoposide and despite its

**Figure 3.** SPRTN is a DNA binding protease. (A) Western blot analysis of GFP-SPRTN expression. GFP-SPRTN and corresponding mutants were immunoprecipitated after 16 hr of transient transfection in 293T-HEK cells. Arrows indicate full-length (FL) or cleaved fragments. (B) Western blot analysis of histone H3 cleavage in cells overexpressing GFP-SPRTN, GFP-SPRTN-E112A or GFP-SPRTN-ΔC. (C) EMSA of full-length recombinant SPRTN or SPRTN-ΔC (a.a. 1–216) and 0.25 uM 6-FAM-39-mer single stranded oligodeoxynucleotide (6-FAM-ssODN) or (D) 0.25 uM 6'FAM-39-mer double stranded oligonucleotide (6-FAM-dsODN) .

The following figure supplement is available for figure 3:

**Figure supplement 1.** SPRTN self-cleaves in vivo.

chromatin recruitment (*Figure 4—figure supplement 1*), consistent with this mutation being catalytically inactive. We next assessed the effect of SPRTN-ΔC overexpression with respect to histone H3 cleavage. Notably, SPRTN-ΔC overexpression does not produce histone H3 cleavage fragments despite the fact that we observe SPRTN-ΔC chromatin recruitment (*Figure 4—figure supplement 1*). This result suggests that SPRTN needs the C-terminus region to either target histone H3 or for DNA binding.

To distinguish between these two possibilities, we asked if SPRTN could directly bind to DNA. We used purified full length SPRTN and a truncated fragment composed of amino acids 1–216 (SPRTN-ΔC$^{1–216}$) and assessed DNA binding using an electrophoretic mobility shift assay (EMSA) that incorporated the use of a fluorescently labeled 39-mer single or double stranded oligodeoxynucleotide (6-FAM-ssODN or 6-FAM-dsODN, respectively). As SPRTN levels were gradually increased, we observed a corresponding increase in the electrophoretic mobility of the DNA probe, indicative of SPRTN binding to single or double stranded DNA (*Figure 3C and D*). The electrophoretic mobility shift is not detected when we employ the SPRTN-ΔC$^{1–216}$ protein in this EMSA assay. This result suggests that a region(s) located in the carboxyl-terminal half of SPRTN contains a DNA binding domain as detected in purified in vitro binding assays and that this domain is probably required to target histone H3 for degradation in cells.

To formally test whether SPRTN is a protease in vitro, we utilized purified recombinant SPRTN in in vitro cleavage reactions. Purified SPRTN elicits auto-cleaving activity in the presence of double stranded DNA similar to that observed in vivo (*Figure 4A*). Importantly, the auto-cleaving activity is dependent on the conserved catalytic glutamic acid residue at position 112, because the SPRTN-E112A mutant is devoid of autocatalytic activity. In addition, SPRTN autocatalytic activity is inhibited by the zinc chelator, 1,10-Phenantroline, indicating that SPRTN is a zinc-dependent metalloprotease. In vitro, the SPRTN-Y117C mutant protein has no detectable autocatalytic activity. In sharp contrast to the SPRTN-Y117C, and to what is observed in cells upon SPRTN-ΔC expression, recombinant SPRTN-ΔC protein is catalytically active as it auto-cleaves into smaller cleavage fragments in in vitro reactions (*Figure 4A*). To further characterize SPRTN as a protease, we showed that purified SPRTN cleaves itself and histone H3 in in vitro settings in a time-dependent manner (*Figure 4B*). Importantly, we did not detect histone H3 cleavage when treated with SPRTN-E112A. Similarly, SPRTN-Y117C also failed to cleave histone H3 at the latest time-point. To test whether SPRTN can directly act on Top2, we used purified SPRTN or SPRTN-E112A and purified Top2 in in vitro cleavage reactions. We found that SPRTN, but not SPRTN-E112A, can reduce the levels of full length Top2. Interestingly, the addition of recombinant ubiquitin can enhance not only Top2 cleavage, but also SPRTN self-cleavage, whereas addition of recombinant SUMO cannot mediate this enhanced effect (*Figure 4C*). To further assess the effect of free ubiquitin on SPRTN self-cleavage, we compared the kinetics of SPRTN self-cleavage, in a time-course experiment, while in the absence or presence of ubiquitin. The addition of DNA induces SPRTN self-cleavage in a time dependent manner, with enhanced SPRTN cleavage fragments seen at later time points. This effect is further enhanced when free ubiquitin is included in the reaction (*Figure 4—figure supplement 2A*). Importantly, the addition of ubiquitin has no effect on SPRTN-E112A or SPRTN-Y117C self-cleavage (*Figure 4—figure supplement 2B*), ruling out the presence of a contaminating protease in our ubiquitin preparations. Taken together, these results clearly show that SPRTN is a zinc-dependent metalloprotease that can proteolyze itself and other DNA binding proteins, and that presence of free ubiquitin in the reaction enhances SPRTN cleavage activity.

## SPRTN-mediated DPC activity is compromised in cells derived from patients with Ruijs-Aalfs syndrome

To gain further molecular insight into the compromised activities of SPRTN-Y117C and SPRTN-ΔC mutant proteins identified in Ruijs-Aalfs syndrome patients, we more closely analyzed the primary amino acid sequence of these mutant proteins. Careful analysis of the primary SPRTN amino acid sequence revealed the presence of at least three potential nuclear localization signals (NLS) (*Figure 5—figure supplement 1*). Of these three, one of them (NLS3) was a stronger NLS candidate for two reasons: (1) The NLS lies between the PIP and UBZ domain and would clearly be missing in the SPRTN-ΔC protein. (2) The NLS amino acid sequence is conserved across higher vertebrate SPRTN orthologues (*Figure 5A*).

To test if the putative NLS is necessary for SPRTN nuclear expression, we transduced U2OS cells with lentiviral particles encoding GFP-SPRTN and corresponding mutant alleles. In agreement with previous reports, GFP-SPRTN expression is confined to the nucleus (*Figure 5B*) (*Centore et al., 2012*; *Davis et al., 2012*; *Ghosal et al., 2012*; *Juhasz et al., 2012*; *Kim et al., 2013*; *Lessel et al., 2014*; *Machida et al., 2012*; *Mosbech et al., 2012*). The SPRTN-Y117C mutation has no effect on localization akin to catalytic inactive SPRTN-E112A mutant. However, the GFP-SPRTN-ΔC protein is evidently mislocalized to the cytoplasm (*Figure 5B*). This apparent SPRTN-ΔC mislocalization



**Figure 4.** In vitro SPRTN substrate cleavage. (**A**) In vitro SPRTN self-cleavage reactions. Purified proteins were incubated with or without DNA for 5 hr at 37°C. 1,10 Phe = 1,10 Phenanthroline, a zinc metal chelator and inhibitor of zinc metalloproteases. Proteins were separated by SDS-PAGE and stained with coommasie blue. (**B**) In vitro histone H3 cleavage. SPRTN was incubated with or without histone H3 (SPRTN:H3 molar ratio of 4:1) in the

*Figure 4 continued on next page*

*Figure 4 continued*

presence of dsDNA for the indicated time points. SPRTN-E112A or SPRTN-Y117C mutants were incubated for 2 hr. Proteins were separated on an SDS-PAGE gel and transferred to a membrane for Western blot analysis. Histone H3 cleavage as well as SPRTN self-cleavage were monitored by immunoblotting with antibodies against histone H3 and SPRTN. (**C**) In vitro Top2 cleavage. Purified recombinant Top2 was pre-incubated with DNA and Etoposide to irreversibly bind Top2 to DNA. Recombinant SPRTN or SPRTN-E112A was then added alone or in combination with 10-fold molar excess of either ubiquitin or SUMO and incubated for 2 hr at 37°C. Proteins were separated by SDS-PAGE and transferred to a membrane for Western blot analysis. Membrane was stained with amido black to detect SPRTN cleavage fragments.

The following figure supplements are available for figure 4:

**Figure supplement 1.** SPRTN recruitment to chromatin.
**Figure supplement 2.** SPRTN activation by ubiquitin.

---

suggests an obvious molecular explanation for compromised function in DPC removal. To further explore the putative NLS sequence in SPRTN, we mutated two residues in SPRTN, namely R408A and L411A and stably transduced U2OS cells with lentivirus encoding GFP-SPRTN-R408A L411A. As expected, GFP-SPRTN-R408A L411A is mislocalized to the cytoplasm similarly to the GFP-SPRTN-Δ C (*Figure 5B*). To explore if the NLS is sufficient for nuclear SPRTN expression, we created a chimeric protein consisting of SPRTN-ΔC and the NLS from the c-myc transcription factor (SPRTN-ΔC +NLS). The NLS fusion to GFP-SPRTN-ΔC is sufficient for restoring nuclear expression (*Figure 5B*) and histone H3 cleavage in cells (*Figure 4—figure supplement 1*). To functionally assess the DPC resolving activity of SPRTN-ΔC+NLS and of SPRTN-R408A L411A, we again employed the ICE assay. As expected, the addition of a heterologous NLS to SPRTN-ΔC, as in *SPRTN-ΔC+NLS*, enhances its Top2 resolvase activity relative to SPRTN-ΔC (*Figure 5C* and *2C*). It then follows, that removal of the NLS signal from full length SPRTN, as in *SPRTN-R408A L411A*, would result in an attenuation of DPC resolvase activity (*Figure 5C*). Thus, we conclude that SPRTN-ΔC is hypomorphic, at least in part, due to its mislocalized expression and its inability to bind to DNA. In addition, we conclude that the NLS in SPRTN is both, necessary and sufficient, for proper nuclear localization and function.

Having already established that *SPRTN-KO* cells are sensitive to DPC-inducing drugs and that SPRTN has proteolytic activity both in vivo and in vitro, we asked if cells derived from patients with Ruijs-Aalfs syndrome were also compromised in DPC resolving activity. To test this hypothesis explicitly, we treated patient derived lymphoblastoid (LCL) cells (B-II-1) with formaldehyde, CPT, or Etoposide and used the pan-DNA damage marker, γ-H2AX, as a proxy for DNA damage resulting from DPC-induced double strand breaks. Control LCLs show background levels of γ-H2AX staining, commonly observed in most cell lines (*Figure 5C*). Compared to control LCLs however, B-II-1 LCLs have more γ-H2AX staining in formaldehyde and Etoposide treatment as the majority of patient cells have increased DNA damage. B-II-1 LCLs contained modest increase in γ-H2AX staining when exposed to CPT treatment. To functionally test the consequence of cells expressing biallelic SPRTN mutant variants, we employed the ICE assay to assess Top2cc removal from chromatin after an acute Etoposide treatment. Consistent with SPRTN-ΔC and SPRTN-Y117C mutant variants being compromised for proteolytic activity, Top2cc are removed from DNA but with slower kinetics relative to control LCLs (*Figure 5D*). Taken together, these results provide a molecular explanation for the defects associated with Ruijs-Aalfs syndrome and shed light on how mutations in SPRTN can contribute to segmental progeroid syndrome, genome instability and hepatocellular carcinoma development (*Figure 6*).

## Discussion

Aging is a phenomenon generally defined as the time-dependent decline in function of cells and organs and it affects all living organisms. DNA damage has emerged as a major perpetrator in aging-related pathologies and cancer. Substantial insights into the link between DNA damage and aging have emerged from studies on human progeroid syndromes. Because the types of DNA

**A.**

| | | | |
|---|---|---|---|
| *H. sapiens* | 402 | DTFPNKRPRLEDKTVF | 413 |
| *B. taurus* | 401 | GKLPSKRPRIEDKTFF | 416 |
| *M. musculus* | 413 | DQFLNKRPRLED---- | 424 |
| *R. norvegicus* | 412 | DQFLNKRPRLED---- | 423 |
| *X. tropicalis* | 192 | YSQKRKRNND------ | 201 |

SPRTN-Y117C

SprT-like    SHP    PIP    NLS    UBZ
1                                        489
▲ Y117C

SPRTN-ΔC

SprT-like
1

**B.**

SPRTN | SPRTN-R408A L411A
SPRTN-ΔC | SPRTN-ΔC +NLS
SPRTN-Y117C | SPRTN-E112A

**C.**

Recovery
Etoposide (hours):   0   0.5   1   2

SPRTN
SPRTN-KO
SPRTN-KO {
SPRTN-WT
SPRTN-ΔC+ NLS
SPRTN-R408A L411A

anti-Top2

SPRTN
SPRTN-KO
SPRTN-KO {
SPRTN-WT
SPRTN-ΔC+ NLS
SPRTN-R408A L411A

anti-DNA

SPRTN-KO +
SPRTN, SPRTN-WT, SPRTN-E112A, SPRTN-ΔC+NLS, SPRTN-R408A R411A

100
75
63
anti-GFP

135
anti-vinculin

**D.**

Control    B-II-1

Formaldehyde

Etoposide    Counts

Camptothecin

γ-H2AX    γ-H2AX

■ 0 hr.
■ 1 hr.
■ 4 hr.

**E.**

Etoposide (hours):   0   0.5   1   2

control LCL
B-II-1
anti-Top2

control LCL
B-II-1
anti-DNA

**Figure 5.** SPRTN mutant proteins from Ruijs-Aalfs syndrome patients are hypomorphic for DPC removal. (**A**) Comparison of SPRTN-Y117C and SPRTN-ΔC mutant proteins found in patients with Ruijs-Aalfs syndrome. The NLS sequence highlights the amino acid conservation in vertebrate SPRTN species. (**B**) Fluorescent expression of GFP-SPRTN and mutant proteins in live U2OS cells. (**C**) ICE assay of DNA isolated from SPRTN control, SPRTN-KO cells or SPRTN-KO cells reconstituted with *SPRTN-WT, SPRTN-ΔC+NLS,* or *SPRTN-R408A L411A.* Cells were either untreated or treated for 30 min
*Figure 5 continued on next page*

Figure 5 continued

with 25 uM Etoposide and immediately subjected to ICE assay or 1 and 2 hr after washing out Etoposide (recovery), as in **Figure 2C**. Western blot indicates SPRTN levels after reconstitution of *SPRTN-KO* MEFS. (**D**) Representative flow cytometry analysis of control LCL cells and patient (B-II-1) LCL cells treated with formaldehyde, Etoposide or CPT for 0, 1 or 4 hr. Cells were collected at indicated time points and processed for γ-H2AX staining. Untreated controls are indicated in black for comparison to treated cells. 10,000 cells were scored for each experiment. (**E**) ICE assay of DNA isolated from control or B-II-1 cells that were either untreated or treated with 25 uM Etoposide for the indicated times as in **Figure 2C** and **5C**.

The following figure supplement is available for figure 5:

**Figure supplement 1.** SPRTN nuclear localization signals.

lesions are diverse, intricate DNA repair and checkpoint systems have evolved to detect and repair specific lesions. Here we have focused on elucidating the molecular mechanisms that lead to genomic instability in cells from patients with Ruijs-Aalfs syndrome that have underlying mutations in the *SPRTN* gene. We identify SPRTN as the first mammalian protease to function as a DNA repair protein that resolves DNA-protein crosslinks. Moreover, SPRTN's proteolytic activity is essential for providing tolerance to DPC-inducing agents. Cells that have undergone mutational ablation of the carboxy-terminal half of SPRTN or single amino acid substitution of a tyrosine to cysteine at position 117 such as those identified in the Ruijs-Aalfs syndrome probands, are compromised for proteolytic activity and are sensitive to DPC-inducing agents. Our work revealed that these mutant alleles are hypomorphic for proteolytic activity for at least two reasons. First, SPRTN-ΔC is missing the critical NLS at the carboxy-terminal domain that makes it necessary for nuclear import and thereby SPRTN-ΔC cannot adequately reach its targets. Second, the C-terminal domain harbors a major DNA binding domain that is missing in SPRTN-ΔC. Despite these shortcomings, the hypomorphic SPRTN-ΔC protein retains enough proteolytic activity to sustain life. Evidently, in vitro SPRTN-ΔC is catalytically active and at least one Ruijs-Aalfs syndrome patient with homozygous *sprtn-ΔC* alleles lived through his adolescent years until eventually succumbing to hepatocellular carcinoma in early adulthood (**Ruijs et al., 2003**). In these patient derived cells, SPRTN-ΔC most probably enters the nucleus of proliferating cells during each prophase – concomitant with nuclear envelope breakdown. Contrary to SPRTN-ΔC expression in humans, complete genetic ablation of *SPRTN* is embryonic lethal in mice. This finding argues that SPRTN's protease activity is essential for cellular viability. Consistent with this notion, conditional *SPRTN* MEFs failed to proliferate when reconstituted with *SPRTN-E112A* or *SPRTN-Y117C* alleles as the only *SPRTN* source. In addition, like the catalytic inactive SPRTN-E112A mutant protein, SPRTN-Y117C has very weak catalytic activity in vitro and in vivo. Lastly, mice with hypomorphic SPRTN alleles display accelerated aging features. Our finding that the *SPRTN-ΔC* and SPRTN-Y117C alleles found in Ruijs-Aalfs syndrome patients are hypomorphic is consistent with accelerated aging phenotypes observed in the hypomorphic SPRTN mouse model - linking DPC repair deficiency to segmental progeroid syndrome.

The conservation of DNA repair mechanism across species underscores the importance of genome maintenance. SPRTN and Wss1 may form a common family of DPC resolving proteases in vivo (**Stingele et al., 2015**). Indeed, SPRTN and Wss1 are functionally related as they are both implicated in the repair of DPCs. Mechanistically, however, differences exist as SPRTN cannot functionally complement Wss1 activity in budding yeast. This inability of SPRTN to complement Wss1 could be attributed, in part, to the differing ubiquitin and SUMO binding abilities that are also reflected in subsequent activation of their respective protease domains. In addition, Wss1 is not essential in yeast, whereas SPRTN is essential in mammals (**Biggins et al., 2001**; **Maskey et al., 2014**; **Winzeler et al., 1999**). The fact that *wss1Δ* yeast cells are resistant to CPT and mildly sensitive to formaldehyde, whereas *SPRTN-KO MEFs* are highly sensitive to CPT and formaldehyde strongly argues that partially overlapping pathways contribute to DPC bypass in yeast (**Stingele et al., 2014**). *SPRTN* is also dispensable in *C. elegans* (**Mosbech et al., 2012**), further underscoring the hypothesis that in lower eukaryotes such as in yeast and worms, other pathways can contribute to resolving DPCs.

There are numerous proteins that are intimately associated with DNA, yet these proteins are spared from indiscriminate SPRTN-dependent proteolysis. Therefore, the activity of SPRTN must be carefully regulated. How SPRTN activity is regulated to target DPCs is still an open question. One



**Figure 6.** Model of SPRTN removal of DNA-protein crosslinks. DNA-protein crosslinks may arise stochastically as a consequence of normal metabolic processes occurring at or in the vicinity of DNA or by exogenous sources such as Etoposide. DPCs are resolved by SPRTN to allow for unperturbed and timely DNA replication. When SPRTN is absent or when its activity is compromised, such as in patients with Ruijs-Aalfs syndrome (SPRTN-Y117C or

*Figure 6 continued on next page*

*Figure 6 continued*

SPRTN-ΔC), DPCs can persist and perturb DNA replication and transcription (not shown). DPCs, if not resolved, can lead to genomic instability, tumorigenesis and contribute to accelerated aging.

potential explanation could be that post-translational modification of SPRTN dictates whether SPRTN can access proteins for proteolysis. Specifically, the mono-ubiquitylation of SPRTN may render it in a closed/inactive conformation. Indeed, we have previously uncovered a mono-ubiquitin-mediated mechanism of inactivation for the translesion polymerase, Pol eta (*Bienko et al., 2010*), setting precedent for this mode of regulation. Moreover, Pol eta and SPRTN are both UBZ-containing proteins and both are mono-ubiquitylated in an UBZ-dependent manner (*Bienko et al., 2005*; *Centore et al., 2012*; *Mosbech et al., 2012*). It is conceivably possible that mono-ubiquitylated SPRTN is inactivated because its active site and/or DNA binding site is occluded akin to how Pol eta mono-ubiquitylation prevents its recruitment to PCNA. Alternatively, in trans binding to other ubiquitylated proteins in the vicinity may lead to allosteric activation resulting in the opening of the protease domain. This model is supported by evidence that the addition of ubiquitin, but not SUMO, promotes SPRTN cleavage of Top2 in vitro. Similarly, binding to PCNA or p97 may contribute to the same phenomena, leading to a step-wise activation process in vivo. Based on these and other data we propose a model where SPRTN's proteolytic activity is restricted by a naturally occurring conformation that occludes the protease active site. Either substrate/DNA binding or de-ubiquitination of SPRTN leads to allosteric changes in SPRTN that expose its active site thus unleashing its proteolytic activity. This mechanism appears different in the case of Wss1, whereby binding to SUMO rather than to ubiquitin may act as a critical signal for Wss1 activation. Clearly more work will be required to understand the molecular mechanism underlying substrate selection for members of the DPC protease family including SPRTN and Wss1.

## Materials and methods

### Yeast strains

W303a yeast cells were used to construct targeted gene deletions using standard yeast methods. To construct an SPRTN expression construct under the endogenous Wss1 promoter, 500 nucleotides upstream of the *WSS1* start site was amplified by PCR from genomic DNA and ligated to a PCR product containing GFP-SPRTN to create pWss1-GFP-SPRTN. This fragment was subsequently subcloned into pRS415 and sequenced verified. To create yeast strains expressing FLAG-SPRTN under the control of the galactose inducible promoter, FLAG-SPRTN was cloned into pDONR223 (Invitrogen, Germany) using the Gateway cloning system and subsequently shuttled into pAG-306GAL-ccdB (pAG-306GAL-ccdB was a gift from Susan Lindquist Addgene plasmid #14139). The plasmid was linearized at the URA3 loci and transformed into yeast strains.

### Spot assays

Yeast cells were spotted essentially as described in *Lopez-Mosqueda et al. (2010)*. Briefly, yeast cells were grown overnight in LEU2 dropout media. 0.2 $OD_{600}$ of cells were collected and diluted in five-fold serial dilutions. Cells were spotted on LEU2 dropout plates containing 2 ug/mL doxycycline and/or 40 uM CPT. Yeast plates were incubated at 30°C for 96 hr. For Galactose-inducible FLAG-SPRTN expression, yeast cells were grown as above before spotting on complete plates with 2% glucose with or without 40 uM CPT or 2% galactose with or without 40 uM CPT.

### Cell lines and antibodies

U2OS, HeLa and 293T-HEK cells were purchased from the American Type Culture Collection (Manassas, VA). Their identities were authenticated by STR analysis. B-II-1 patient LCL and control LCL cells were provided by Kristijan Ramadan (Cancer Research UK, Oxford UK). H7 conditional *SPRTN* MEFs were provided by Yuichi Machida (Mayo Clinic, Rochester Mn.) All cells were confirmed to be mycoplasma-free by PCR detection of mycoplasma (Minerva Biolobs, Germany). Human SPRTN was cloned into pEGFP-C1 (Clonetech, France) and point mutations were introduced by site-directed

mutagenesis (Agilent, Germany). SPRTN was cloned into pDONR223 (Invitrogen) and subsequently shuttled into pHAGE-GFP-Blasticidin lentiviral vector using the Gateway System (Invitrogen). Lentiviral particles were produced in 293T HEK cells by transfecting with Genejuice transfection reagent (EMD Millipore, Germany) the lentiviral plasmid encoding SPRTN, pMD2.G, and psPAX2 packaging plasmids (pMD2.G and psPAX2 were a gift from Didier Trono Addgene plasmids #12259 and #12260). Media containing lentivirus was collected 72 hr post-transfection. H7 conditional *SPRTN* MEFs were treated with 4-OHT for 24 hr and media was replaced. At 36 hr post 4-OHT, H7 MEFS were transiently transfected with pHAGE-GFP-SPRTN-E112A or pHAGE-GFP-SPRTN-Y117C using GenJet In vitro DNA transfection reagent (SignaGen Laboratories- Rockvilled, MD.) in 10 cm dish format as described in manufacturers instructions. GFP positive cells were sorted by Flow cytometry. U2OS cells were transduced for 24 hr and selected for 10 days in Blasticidin. GFP-SPRTN-mCherry was cloned by PCR amplifying mCherry from pMcherry-N1 (Clonetech). The PCR product was cloned downstream of GFP-SPRTN (pEGFP-C1) and the plasmid sequence was verified. HeLa cells were transfected with GFP-SPRTN-mCherry using Genejuice (EMD Millipore). The following antibodies were used in this study: Anti-SPRTN (Sigma, Germany; HPA025073), anti-SPRTN (Sigma; SAM1409875) Anti-HA (Santa Cruz, Germany; sc-57592), anti-γ-H2AX (Millipore, Germany; 05–636), anti-GFP (Santa Cruz; sc-9996), anti-Histone H3 (Abcam, Cambridge, UK; Ab1791), anti-Thymine dimers (Kamiya Biomedical, Seattle, WA.; MC-062), anti-FLAG (Sigma; F3165), anti-Topoisomerase 2-alpha (Abcam; ab52934), and Alexa647 Donkey anti-mouse fluorescent secondary antibody (Jackson ImmunoResearch Labs, West Grove, PA.; 715-605-151).

## In vivo complex of enzyme (ICE) assay

Etoposide (Santa Cruz; CAS 33419-42-0) was dissolved in DMSO to a 50 mM stock concentration. Cells were treated with 25 uM Etoposide to induced Top2 DPCs and were detected essentially as described in *Nitiss et al. (2012)*, without modification.

## Clonogenic assay

500 cells were seeded in triplicate in 6-well plates. 24 hr after seeding, cells were treated with Etoposide (72 hr), Camptothecin (Santa Cruz; CAS 7689-03-4)(12 hr) and formaldehyde (Sigma; F8775) (12 hr) before washing out the drug and replacing cells with DMEM + FBS. Cells were allowed to grow for a total of 10 days before fixing with 4% paraformaldehyde and staining with cresyl violet. Colonies with more than 50 cells were scored. Surviving fractions were calculated and plotted. Data represents the mean of three replicate experiments. Survival fractions were calculated as [number of colonies formed after treatment / (number of cells seeded X PE)]. PE = plating efficiency and is calculated as (number of colonies formed/ number of cells seeded) X 100%. Camptothecin was dissolved in DMSO at 25 mM stock concentration.

## Recombinant SPRTN expression and purification

For overexpression in *E.coli* cells, SPRTN constructs were cloned into either the pNIC-ZB vector (full length SPRTN-WT, SPRTN-E112A, and SPRTN-Y117C), or pNIC28-Bsa4 vector (all other truncated constructs). For purification of full length constructs containing a TEV cleavable Z-basic-his tag, cell pellets were thawed and resuspended in lysis buffer (100 mM HEPES pH 7.5, 500 mM NaCl, 10% glycerol, 10 mM imidazole, 1 mM Tris (2-carboxyethyl) phosphene (TCEP), 0.1% DDM, 1 mM MgCl, 1 x set III protease inhibitors (Merck, Germany). Cells were lysed by sonication and 1 unit of Benzonase was added to lysates before the cell were debris pelleted by centrifugation. Lysates were applied to a Ni-sepharose IMAC gravity flow column, washed with 2 column volumes of wash buffer (50 mM HEPES pH 7.5, 500 mM NaCl, 10% glycerol, 45 mM imidazole, 1 mM TCEP), and eluted in elution buffer (50 mM HEPES pH 7.5, 500 mM NaCl, 10% glycerol, 300 mM imidazole, 1 mM TCEP). Elution fractions were applied directly to a 5 ml HiTrap SP HP column (GE healthcare, Germany), washed with wash buffer (50 mM HEPES pH 7.5, 500 mM NaCl, 1 mM TCEP) and eluted with elution buffer (50 mM HEPES pH 7.5, 1 M NaCl, 1 mM TCEP). The purification tag was cleaved with the addition of 1:20 mass ratio of His-tagged TEV protease during overnight dialysis into buffer A (20 mM HEPES, pH 7.5, 500 mM NaCl, 0.5 mM TCEP). Samples were concentrated by ultrafiltration using a 30 kDa molecular weight cut off centrifugal concentrator and loaded on to size exclusion chromatography using a HiLoad 16/60 Superdex 200 column at 1 ml/min in buffer A. The same

protocol was used for purification of the truncated constructs with the exception of the omission of the SP HP column and the inclusion of a Ni-sepharose rebind following TEV cleavage (to remove his-tagged TEV protease).

### Western blots and chromatin fractionations

Yeast cells were lysed by boiling 6 $OD_{600}$ of cells in 200 uL of pre-heated SDS-sample buffer. Mammalian cells were lysed in 50 mM Tris-HCl pH 7.5, 150 mM NaCl, Complete protease inhibitors (EDTA-free Roche, Germany), 1% NP40, 5 mM BME. For immunoprecipitation of GFP fusion proteins, cells were lysed in 50 mM Tris-HCl pH 7.5, 150 mM NaCl, Complete protease inhibitors (EDTA-free Roche), 1% NP40, 5 mM BME. GFP-trap beads (ChromoTek, Germany) were used per manufacturers instructions, to precipitate GFP-SPRTN. Chromatin fractionations were done essentially as describe in *Mendez and Stillman (2000)*.

### In vitro SPRTN cleavage assays

1.8 uM SPRTN was incubated in the presence or absence of dsDNA (10 uM) and/or ubiquitin (20 uM) in 40 ul of final reaction buffer (25 Tris pH 7.5, 80 mM NaCl) at 37°C. 10 ul aliquots were removed from the reaction mix at 30 min, 1 hr and 2 hr. SDS loading dye was added to stop the reactions. The samples were separated by SDS-PAGE and analyzed by Coommassie staining and Western blot analysis with an antibody against SPRTN.

### In vitro cleavage of histone H3

2 uM SPRTN was incubated with Histone H3 at a molar ratio of 4:1 or 1:1 (SPRTN:H3) in a reaction buffer containing 25 mM Tris pH 7.5 and 80 mM NaCl. dsDNA and ubiquitin were used at concentrations of 10 uM and 20 uM, respectively. The reaction was incubated for 2 hr at 37°C and stopped by the addition of Laemmli buffer. Samples were separated on an SDS-PAGE and Western blot analysis with antibodies against histone H3 and SPRTN

### In vitro cleavage of top2

The reactions were performed in 25 mM Tris pH 7.5, 100 mM NaCl. Top2 was crosslinked to dsDNA using 2 mM Etoposide and incubated prior the cleavage assay. Ubiquitin and SUMO2 were used approximately at 10 times molar excess of SPRTN. A final dsDNA (40 bp long) concentration of 10 µM was used. Reactions were incubated at 37°C for 2 hr and stopped by the addition of 5X Laemmli buffer. Samples were run on an SDS gel followed by Western blotting with Top2 and SPRTN specific antibodies.

### Flow cytometry

Cells were washed 2x with ice cold PBS before fixing in 75% ethanol for 20 min at 4°C. Cells were washed 2x in PBS and blocked in 4% BSA/ 0.1% Tween 20 for 1 hr at room temperature. Cells were stained with antibodies against γ-H2AX and Alexa 647 fluorescent secondary antibody. Single cells were gated using forward scatter and side scatter modes. Cells without γ-H2AX antibody were used as antibody controls. FlowJo software was used for data analysis.

### Electrophoretic mobility shift assay

For analysis of DNA binding, electrophoretic mobility shift assay was used. The respective SPRTN proteins were incubated in 30 uL reactions (10 mM Tris-HCl pH 7.5, 0.2 mM DTT, 5 uM ZnSo4) with 0.25 uM fluorescently labeled ssDNA (6-FAM-ssODN 5'GCGCGCCCATTGATACTAAATTCAAGGA TGACTTATTTC). To generate a dsDNA 6-FAM-ODN, the above oligo was annealed to a reverse complement oligo (GAAATAAGTCATCCTTGAATTTAGTATCAATGGGCGCGC). After a 30 min incubation at 20°C, protein-DNA complexes were separated on 1.5% Agarose gel and visualised by FUSION-SL imager (Vilber, Germany).

### Microscopy

Imaging of live U2OS, or HeLA cells, grown in 8-well Lab-Tek chambers, was performed on Visitron confocal spinning disc microscope, equipped with Yokogawa CSU-X1 Scanning head CSU X1-A1-

5000 rpm, single filter sets for GFP, mCherry, EBFP2 and HXP 120 Xenon Lamp for epi-fluorescence illumination. Still images were processed using ImageJ (NIH).

## Acknowledgements

We would like to thank Yuichi Machida for providing the conditional *SPRTN* MEFs (H7 line). We are grateful to Kristijan Ramadan and Bruno Vaz for providing SPRTN enzymes and help with in vitro cleavage reactions. We are grateful to Brian Luke (IMB-Mainz), Susan Lindquist and Didier Trono for providing plasmids. We are grateful to Hector Lopez for artwork. We thank Dikic lab members for their continued support and constructive discussions. JLM is supported by a post-doctoral fellowship from the Human Frontiers Science Program. This work was supported by grants from the DFG (SFB1177), the Cluster of Excellence 'Macromolecular Complexes' of the Goethe University Frankfurt (EXC115), LOEWE grant Ub-Net and LOEWE Centrum for Gene and Cell Therapy Frankfurt, and Medical Research Council-UK programme grant (MC_PC_12001/1)

## Additional information

### Competing interests

ID: Senior editor, *eLife*. The other authors declare that no competing interests exist.

### Funding

| Funder | Grant reference number | Author |
| --- | --- | --- |
| Deutsche Forschungsge-meinschaft | SFB1177 | Ivan Dikic |
| Deutsche Forschungsge-meinschaft | CEF-MC | Ivan Dikic |
| Human Frontiers Science Program | Postdoctoral fellowship | Jaime Lopez-Mosqueda |
| LOEWE Zentrum CGT and Loewe Network Ub Net | Fellowships | Ivan Dikic |

The funders had no role in study design, data collection and interpretation, or the decision to submit the work for publication.

### Author contributions

JL-M, Conceived the project and wrote the manuscript with input from all other co-authors, Performed yeast and mammalian cell experiments, and nuclear localization studies, Analysis and interpretation of data; KM, Constructed yeast strains and performed EMSA experiments, as well as Wss1 self-cleavage studies, Analysis and interpretation of data; SP, Performed chromatin fractionation assays and have help in mammalian cell work, Analysis and interpretation of data; SK, Performed in vitro cleavage assays with SPRTN and also Top2 assays, Analysis and interpretation of data; IM-T, Performed in vitro cleavage assays, Analysis and interpretation of data; JT, Conception and design, Analysis and interpretation of data, Drafting or revising the article; ID, Conceived the project, Supervized the performance and wrote the manuscript with input from all other co-authors, Analysis and interpretation of data

### Author ORCIDs

Jaime Lopez-Mosqueda, http://orcid.org/0000-0003-0301-1971
Ivan Dikic, http://orcid.org/0000-0001-8156-9511

## Additional files

### Supplementary files

• Supplemental file 1. Yeast strains used in this study.

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
