## [Decision Letter]

Thank you for submitting your article "SPRTN is a mammalian DNA-binding metalloprotease that resolves DNA-protein crosslinks" for consideration by *eLife*. Your article has been favorably evaluated by Jonathan Cooper (Senior Editor) and three reviewers, one of whom, Wade Harper, is a member of our Board of Reviewing Editors. The reviewers have opted to remain anonymous.

The reviewers have discussed the reviews with one another and the Reviewing Editor has drafted this decision to help you prepare a revised submission.

Summary:

The paper is timely and makes several important points, including demonstrating the in vitro protease activity of SPRTN, showing that cells defective in SPRTN fail to remove Top2 from DNA, and demonstrating that mutations in Rujis-Aalfs syndrome are pathogenic. The paper also suggests roles for UB or SUMO in SPRTN activation in vitro, although the mechanisms of this isn't fully developed. This is a rapidly emerging area of DNA repair, and the paper goes some distance towards understanding the function of SPRTN. Nevertheless, the reviewers suggested improvements to the paper, including addition of missing controls and other relevant data.

Essential revisions:

1) The experiments looking at the effect of UB and SUMO effects on activity could be improved and better controlled. This is especially the case if you want to be able to suggest an underlying mechanism. The pattern of cleavage of Wss1 is different when SUMO is added relative to addition of ssDNA. Do you have any info on the different cleavage patterns? Minimally, to be convincing, the input lanes would have to be shown without the Wss1 or better yet with catalytically dead Wss1.

2) For completeness, the authors need to show the levels of SPRTN over-expression in experiments where SPRTN is over-expressed.

3) Additional evidence that SPRTN auto-cleaved in vivo (when over-expressed) – rather than being cleaved post-lysis – would strengthen the paper. The fact that the mutant protein doesn't do this is a reasonable argument, but if you have data on this to include (for example a pulse chase), it would improve the paper.

4) The experiments looking at Wss1 activation by ssDNA are interesting but bring up the question of whether dsDNA also activates, particularly since the SPRTN EMSA assays were done with dsDNA (See subsection “SPRTN is a DNA binding protease in vivo and in vitro”, last paragraph and subsection “in vitro cleavage of Top2”)? This is also relevant to the experiment in Figure 4, since the extract presumably already contains DNA. The reviewers feel the paper would be strengthened by testing both in vitro binding and activation with both ssDNA and dsDNA (and possibly other types of DNA such as forks if available). If SPRTN only binds ssDNA but is activated in vitro only by dsDNA then that wouldn't fit your model and would suggest that there may be some sort of contaminant in the protein preps. These experiments would bridge what appears to be a disconnect between the protease assays and the EMSA.

---

## [Author Response]

[…]

*Essential revisions:*

*1) The experiments looking at the effect of UB and SUMO effects on activity could be improved and better controlled. This is especially the case if you want to be able to suggest an underlying mechanism. The pattern of cleavage of Wss1 is different when SUMO is added relative to addition of ssDNA. Do you have any info on the different cleavage patterns? Minimally, to be convincing, the input lanes would have to be shown without the Wss1 or better yet with catalytically dead Wss1.*

We have included new experiments (Figure 4—figure supplement 1) where we included a time course analysis to compare the kinetics of SPRTN self-cleavage with and without ubiquitin (Figure 4—figure supplement 1). In addition, we include as controls both SPRTN-E112A and SPRTN-Y117C (Figure 4—figure supplement 1).

We have removed the Wss1 self-cleavage assay in the presence of free sumo or free ubiquitin. After repeating this assay with more controls, we found that sumo enhances Wss1 self-cleavage as we initially observed (lane 6), but unfortunately, we also see this sumo-dependent effect with the Wss1 EQ catalytic mutant (lane 7) and in the negative control (lane 11- presence of EDTA). Finally, the use of commercial sumo had no effect on Wss1 self-cleavage (not shown).

Author response image 1.**DOI:**
http://dx.doi.org/10.7554/eLife.21491.015

*2) For completeness, the authors need to show the levels of SPRTN over-expression in experiments where SPRTN is over-expressed.*

Western blots have been included in experiments where SPRTN is over-expressed.

*3) Additional evidence that SPRTN auto-cleaved* in vivo *(when over-expressed) – rather than being cleaved post-lysis – would strengthen the paper. The fact that the mutant protein doesn't do this is a reasonable argument, but if you have data on this to include (for example a pulse chase), it would improve the paper.*

To include additional evidence to support that SPRTN self-cleaves in vivo when over-expressed, we did two additional experiment (Figure 3—figure supplement 1). First, we over-expressed GFP-SPRTN or GFP-SPRTN-E112A in 293T-HEK cells and lysed cells directly in SDS-PAGE sample buffer. This experiment shows the presence of SPRTN cleavage fragments in two independent transfections, but not in the SPRTN-E112A samples. Secondly, we made an SPRTN expression plasmid containing two fluorescent tags – GFP at the amino-terminus and mCherry at the carboxy-terminus (Figure 3—figure supplement 1). Based on the size of the cleavage fragments we observe, we reasoned that SPRTN self-cleaves, probably in trans, in a region just C-terminal to the Sprt-like protease domain. The self-cleavage of GFP-SPRTN would be reminiscent of GFP-SPRTN∆C expression – which is entirely cytoplasmic. Upon over-expression of this plasmid in HeLa cells, we can readily detect GFP fluorescent signal in the nucleus as well as is the cytoplasm, while the mCherry fluorescent signal remains in the nucleus. Taken together, these findings strongly support the notion that SPRTN self-cleaves in cells and is not simply a consequence of in trans cleavage in lysates.

*4) The experiments looking at Wss1 activation by ssDNA are interesting but bring up the question of whether dsDNA also activates, particularly since the SPRTN EMSA assays were done with dsDNA (See subsection “SPRTN is a DNA binding protease* in vivo *and in vitro”, last paragraph and subsection “in vitro cleavage of Top2”)? This is also relevant to the experiment in Figure 4, since the extract presumably already contains DNA. The reviewers feel the paper would be strengthened by testing both in vitro binding and activation with both ssDNA and dsDNA (and possibly other types of DNA such as forks if available). If SPRTN only binds ssDNA but is activated in vitro only by dsDNA then that wouldn't fit your model and would suggest that there may be some sort of contaminant in the protein preps. These experiments would bridge what appears to be a disconnect between the protease assays and the EMSA.*

We have now done SPRTN-DNA binding experiments using the EMSA with either ssDNA or dsDNA, as requested by reviewers. We found that SPRTN can bind to either ssDNA or to dsDNA.